# Anti-Cancer Potential of *Oxialis obtriangulata* in Pancreatic Cancer Cell through Regulation of the ERK/Src/STAT3-Mediated Pathway

**DOI:** 10.3390/molecules25102301

**Published:** 2020-05-14

**Authors:** Eun-Jin An, Yumi Kim, Seung-Hyeon Lee, Hyun Min Ko, Won-Seok Chung, Hyeung-Jin Jang

**Affiliations:** 1College of Korean Medicine, Kyung Hee University, 26, Kyungheedae-ro, Dongdaemun-gu, Seoul 02447, Korea; aej3866@naver.com (E.-J.A.); yumi0201@khu.ac.kr (Y.K.); skyking27@naver.com (S.-H.L.); rhgusals93@naver.com (H.M.K.); 2Department of Science in Korean Medicine, Graduate School, Kyung Hee University, Seoul 02447, Korea; 3College of Korean Medicine and College of Pharmacy, Kyung Hee University, 26, Kyungheedae-ro, Dongdaemun-gu, Seoul 02447, Korea

**Keywords:** *Oxalis obtriangulata*, pancreatic cancer, BxPC3, anti-cancer effect

## Abstract

As a plant medicine, *Oxalidaceae* has been used to treat various diseases in Korea. However, there is little data on the anti-cancer efficacy of *Oxalidaceae*, particularly *O. obtriangulata*. This study aimed to investigate the anti-cancer effect of *O. obtriangulata* methanol extract (OOE) and its regulatory actions on pancreatic carcinoma. OOE showed anti-proliferative effects and induced cell death in the colony formation and cell viability assays, respectively. The Fluorescence-activated cell sorting (FACS) data confirmed that OOE significantly induced cell cycle accumulation at the G2/M phase and apoptotic effects. Additionally, OOE inhibited the activated ERK (extracellular-signal-regulated kinase)/Src (Proto-oncogene tyrosine-protein kinase Src)/STAT3 (signal transducers and activators of transcription 3) pathways including nuclear translocation of STAT3. Furthermore, suppression of Ki67, PARP(Poly ADP-ribose polymerase), caspase-3, P27(Cyclin-dependent kinase inhibitor 1B), and c-Myc as well as the STAT3 target genes CDK(cyclin-dependent kinase)1, CDK2, Cyclin B1, VEGF-1(vascular endothelial growth factor-1), MMP-9(Matrix metallopeptidase 9), and Survivin by OOE was observed in BxPC3. We speculate that these molecular actions might support an anti-cancer effect of OOE. In this study, we demonstrated that OOE may be a promising anti-cancer material and may serve as a natural therapy and alternative remedy for pancreatic cancer treatment.

## 1. Introduction

Pancreatic cancer is one of the most fatal and aggressive cancers and has a particularly low chance of survival. In fact, in the United States, the 5-year survival rate of patients diagnosed with pancreatic cancer is only 8%. Pancreatic cancers are very difficult to diagnose early and treat effectively because of their silent initial symptoms [1]. Typically, the most problematic type of pancreatic cancer is pancreatic ductal adenocarcinoma (PDAC), which occurs in the duct area. Following lung cancer, PDAC is expected to be the second leading cause of cancer-related deaths [2].

Unlike healthy cells, cancer cells have specific abilities that contribute to tumor formation and further cancer progression such as proliferation without growth factors, unlimited replication, evasion of apoptosis, and metastasis [3]. Previous studies have reported that aberrant activation of signal transducers and activators of transcription 3 (STAT3) is involved in oncogenesis in diverse types of human cancer including pancreatic cancer cells [4]. STAT3 phosphorylation can be activated by mutation of the Src protein, and phosphorylated STAT3 is associated with ERK activation. Moreover, because it is a transcription factor that regulates cellular proliferation, invasion, and migration, which are all critical for cancer progression, targeting STAT3 is an appealing anti-cancer strategy for inhibiting cellular oncogenic functions [5].

The pharmaceutical industry has produced chemotherapy drugs that target and kill cancer cells using their unique characteristics compared to normal cells. However, such drugs are limited because they cause toxicity such as cardiotoxicity and myelotoxicity in the human body. Furthermore, radiation therapy accompanied by chemotherapeutic drugs is used to suppress the cellular replication system in cancer cells, but it has the problem of low selectivity [6]. Thus, currently used treatments for cancer using these approaches have some obstacles in considering toxicity and drug resistance. In this situation, plants can be a safe and promising resource against cancer cells [7]. Herbal-derived drugs such as polyphenols and taxols are already available natural medicines for cancer therapies. Some researchers have also suggested the conjugation of alternative treatments and conventional therapy to more effectively impact cancers. Plants with a high potential to provide newer drugs as anti-cancer agents can be an effective alternative treatment [8].

*Oxalis obtriangulata* is a plant belonging to the *Oxalidaceae* family, which is distributed in deep mountains throughout China, Korea, and Japan. The *Oxalidaceae* family is characterized by a sour taste and is a long-established medicinal plant and is known to contain oxalic acid, malic acid, and tartaric acid [9]. In Korea, its leaves and stems have been used for skin diseases such as atopic dermatitis by spraying the juice on the symptomatic area. It has been reported that medicinal herbs containing *Oxalis* reduced acne caused by *Propionibaterium acnes* in mice [10]. In addition, it is reported that *Oxalidaceae* has a detoxifying effect, and as a plant medicine, it has been used to treat various diseases including diarrhea, dysentery, jaundice, and prolapse.

However, there is little data on the anti-cancer efficacy of *Oxalidaceae*. The present study aimed to examine the effect of the methanol extract of *Oxalis obtriangulata* on ERK/Src/STAT3 activation in a human PDAC cell line, BxPC3, by assessing the induction of apoptosis, cell cycle arrest, and anti-proliferative effects. These data will provide novel insight regarding *Oxalis obtriangulata* as a natural source of anti-cancer agents against pancreatic cancers.

## 2. Results

### 2.1. O. Obtriangulata Methanol Extract (OOE) Affected Pancreatic Cell Viability

The MTT assay was conducted to elucidate the cytotoxicity of OOE on BxPC-3, AsPC-1, and MIAPaCa-2 pancreatic cancer cells. OOE at 12.5–400 μg/mL significantly decreased cell viability in a concentration-dependent manner in three cells. Among them, OOE showed powerful inhibition of cell viability in BxPC-3 cells (Figure 1). In A549 (human lung carcinoma), HepG2(human liver carcinoma), and GES-1(human gastric normal cell), OOE also showed growth inhibitory effect (Appendix A). However, at the same concentration (200–400 μg/mL), cytotoxicity was highest in BxPC3. Based on this result, we aimed to study the anti-cancer potential of OOE on BxPC3 cells.

### 2.2. OOE Affected Cell Proliferation

To determine the long-term effects of OOE on cell proliferation, the colony formation assay was performed for 14 days. As shown in Figure 2A, OOE (50–200 μg/mL) inhibited cell colony formation effectively when compared with the control. The stained plate wells were measured for intensity, as shown in Figure 2B. These findings were also supported by the inhibitory effect of OOE on the expression of Ki67, a cell proliferation marker, in BxPC3 nuclei (Figure 2C,D). These data indicated that OOE effectively inhibited BxPC3 cell proliferation and induced cell death.

### 2.3. OOE Arrested the Cell Cycle at G2/M Phase and Induced Apoptotic Effects in BxPC3 Cells

The effects of OOE on inducing apoptosis and cell cycle arrest were analyzed by flow cytometry. While the G0/G1 phase peak was decreased, BxPC3 cells treated with OOE (100 and 200 μg/mL) showed an accumulation at G2/M phase (Figure 3A). Cells treated with 200 μg/mL OOE exhibited significantly increased G2/M phase arrest from 21.9% to 67.6% and decreased G0/G1 phase accumulation from 57.6% to 16.5% (Figure 3C). The FACS results showed that OOE triggered apoptosis in BxPC3 cells in a dose-dependent manner (Figure 3B). The percentage of total apoptosis rate was 6% in the control cells. OOE-treated cells showed an increased apoptosis rate compared with the control cells. The percentage of apoptotic cells was 11.55% in cells treated with 100 μg/mL of OOE, and 24.9% in cells treated with 200 μg/mL of OOE (Figure 3D). Based on the above experimental results, OOE can effectively induce G2/M phase arrest and apoptotic effect in BxPC3 cells.

### 2.4. OOE Inhibited Phosphorylation of the ERK/Src/STAT3 Signaling Pathway in BxPC3 Cells

As a transcription factor, STAT3 is an important regulator in various cancer cells including pancreatic cancer cells. Src kinase is well defined as a STAT3 activator. Historically, the ERK family of mitogen-activated protein (MAP) kinases also plays an important role in cancer proliferation and tumor phenotype. To determine whether OOE can inhibit the activation of ERK/Src/STAT3 pathways, western blotting was employed to confirm phosphorylated protein expression levels. OOE significantly dose-dependently inhibited the expression of *p*-Src, *p*-ERK, and *p*-STAT3 (both at Tyr705 and Ser727) (Figure 4A–C). Furthermore, STAT3 detection in nuclei showed that OOE also reduced nuclear translocation of STAT3, which promotes the expression of downstream genes related with the cell cycle, apoptosis, and metastasis (Figure 4D). This was also supported by immunofluorescence data. As shown in Figure 4E,F 200 μg/mL OOE significantly decreased nuclear translocation of STAT3 compared with the control group.

### 2.5. OOE Downregulated Various STAT3 Target Genes, Which Are Related with Cell Apoptosis, Proliferation, and Cell Cycle Arrest in BxPC3 Cells

Western blot analysis showed that OOE-treated cells exhibited increased cleaved PARP and caspase-3 protein levels compared with the control (Figure 5A). In line with Figure 2 and Figure 3, OOE also showed downregulated protein expression levels of Ki67 and c-Myc, inducing activation of p27 (Figure 5B). Compared to the control, OOE reduced the mRNA levels of cyclin B1, cyclin-dependent kinase 1 (CDK1), and CDK2, which are related to cell cycle arrest (Figure 5C–E), as well as those of cell survival and proliferation-related factors such as Survivin, VEGF-1, and MMP-9 (Figure 5F–H).

### 2.6. LC-MS Analysis of OOE

We analyzed components in OOE to investigate which components play a major role in its anti-cancer activities. The oxalic acid, malic acid, and tartaric acid, which are commonly known as components of the Oxalis family, were found to be present in a small amount in OOE (Appendix A). Since a complete component investigation of *O. obtriangulata* has yet to be reported, we only confirmed that other OOE components can be separated, as shown in Figure 6. Although the reported studies were insufficient for identifying the components by matching peaks and molecular weights, we used Qualitative Analysis Software to predict the molecular weights of numbered peaks, as depicted in Figure 5 and Table 1.

## 3. Discussion

Pancreatic ductal adenocarcinoma (PDAC) has proven to be among the most unbending targets in the modern era in terms of cancer treatment. Achieving a breakthrough with pancreatic cancer is difficult because early diagnosis is not possible, diagnosis is inaccurate, and the few treatments are burdened with many patients. Therefore, it is necessary to identify efficacious and stable substances for treating PDAC [2]. This research was performed to distinguish OOE as a novel source of potential anti-cancer agents based on the blockage of the STAT3 pathway in a human pancreatic ductal cancer cell line, BxPC3. Our data comprehensively showed that OOE inhibited proliferation, induced cell cycle arrest, and had a slight apoptotic effect in BxPC3 cells. OOE modulated the expression of ERK, Src, and STAT3, which are upregulated in BxPC3 cells as well as STAT3-targeted downstream genes including Cyclin B1, CDK1, CDK2, PARP, caspase-3, MMP-9, VEGF-1, Ki67, p27, c-Myc, and Survivin.

STAT3 is a signaling molecule that delivers signals detected, along with Src activation, through growth factors or cytokine receptors from the cell membrane to the nucleus [11]. Previous study has shown that STAT3 plays a critical role in cell apoptosis, metastasis, proliferation, and immunomodulation in cancer cells including pancreatic cancer [12]. For its activation, STAT3 has two important phosphorylation sites, Tyr705 and Ser727. In PDAC, constitutive phosphorylation of STAT3 at Tyr705 has been reported in 30–100% of human tumor specimens. Additionally, phosphorylation at Ser727 has been considered as a secondary event after Tyr705 phosphorylation [13]. The functional differences between the two phosphorylation sites in pancreatic cancer have not been clearly identified. However, one study has shown that Ser727 phosphorylation is involved in cell survival and nuclear translocation of STAT3 regardless of Tyr705 phosphorylation in melanoma cells [14]. In our study, OOE suppressed Src and STAT3 phosphorylation (at both Tyr705 and Ser727) as well as STAT3 transcriptional action through inhibiting nuclear translocation. Moreover, we speculated that the blockage of both phosphorylation sites may be due to the anti-cancer potential of OOE. Tyrosine phosphorylation of STAT3 can be stimulated by cytosolic kinases including Src, and the secondary phosphorylation of serine is modulated by ERK [15]. The secondary phosphorylation is considered to enhance the capacity of tyrosine-phosphorylated STAT3, thus promoting the transcription of cell function-related factors in cancer. ERK also widely affects diverse cell processes including not only cancer proliferation, but also metastatic and angiogenic effects [16]. Moreover, a previous study reported that in BxPC3 cells, chemotherapeutic agents such as gemcitabine could induce ERK1/2 activity [17]. Hence, this ERK1/2 activation seriously contributes to chemotherapy resistance in pancreatic cells [17]. In connection with this, it is also reported that the silencing of pERK1/2 sensitizes BxPC3 cells to gemcitabine-induced apoptosis [18]. Accordingly, the suppressive effect of OOE on ERK indicates that OOE not only inhibits BxPC3 cell proliferation and growth, but also has a possibility of lowering activated ERK by chemotherapy such as gemcitabine (Appendix A). Considering this, OOE may be an appropriate alternative treatment to enhance chemotherapy.

STAT3 coordinates the expression of genes involved in cell cycle regulation. Among them, CDK1 and the cyclin B1 complex are key modulators of the G2/M phase checkpoint [19]. CDK2, which is well known as an S phase marker, is also reported to contribute to G2/M progression by facilitating cyclin B accumulation [20]. Problems with the G2/M phase due to defects in these genes may cause damaged cells to enter mitosis and undergo apoptosis, therefore increasing the cytotoxicity of chemotherapy.

Activated c-Myc is a hallmark of cancers, which is associated with most human tumor types. c-Myc activation is involved in the growth of cancer cells including transcription and replication of DNA, and regulation of stemness and differentiation of cancer [21]. One of the targets inhibited by myc is the P27 CDK inhibitor, which plays a major role in the cell cycle process. P27 is a tumor suppressor that binds to the cyclin-cdk complex to repress the activity and mitogenic signals in cancer cells. Decreased levels of P27 are related with tumor grade and progression stage in various human carcinoma including colorectal and breast cancers. In oncogenic cells, P27 inhibited the cell proliferation. Moreover, tumor development is prevented by P27 activation suppressing cell cycle process [22,23]. In this study, OOE inhibited the mRNA expression levels of CDK1, CDK2, and cyclin B1. This indicated that OOE could induce G2/M phase arrest starting from the S phase, which is consistent with our cell cycle analysis data. The arrest effect of OOE also reduced the expression of Ki67, P27, and c-Myc, the markers of proliferation and cell cycle arrest, in BxPC3. Taken together, these data suggest that OOE induced delayed proliferation with potent cell cycle arrest through the inhibition of these multiple factors.

Meanwhile, the apoptotic pathway is another target in pancreatic cancer, and caspase-3 is a central effector caspase that initiates apoptosis signals. Activated caspase-3 cleaves PARP protein, and cleaved PARP is considered to be a hallmark of apoptosis [24]. According to our data, OOE showed increased cleaved caspase-3, cleaved PARP, and mRNA expression of Survivin, one of the apoptotic downstream genes of STAT3. OOE also inhibited the expression of metastasis-related factors such as MMP-9 and VEGF, which are targets of STAT3. The FACS data showed OOE significantly increased apoptosis rate. This result may be due to the inclusion of substances with other unexpected effects in OOE, and it supports the need for a component analysis study on OOE. As shown in Figure 6 and Table 1, we performed LC-MS to analyze OOE components and recorded the molecular weights of the substances with matched retention times. However, there is little previous analytical data on OOE to determine what components the peaks represent; thus, additional analytical research is required to confirm and identify the major compound that mainly contributed to the anti-cancer effect of OOE.

In this study, we confirmed the anti-cancer activity of OOE on BxPC3, a pancreatic cancer cell. OOE modulated ERK/Src/STAT3 activation and regulated STAT3-downstream genes related with tumor development. Moreover, OOE affected cell viability, proliferation, and induced apoptotic effect and accumulation at G2/M phase in BxPC3. Considering our results comprehensively, we showed the possibility of OOE as an anti-cancer agent. Additionally, this report suggests a new insight of OOE as a source of potential anti-cancer compound. It is necessary to carry out additional experiments to accurately analyze the components in OOE and evaluate the efficacy of each compound.

## 4. Materials and Methods

### 4.1. Cell Culture

BxPC3, MIAPaCa2, AsPC1, and GES-1 were cultured in Roswell Park Memorial Institute 1640 (RPMI-1640) (Corning Inc., New York, NY, USA). A549 and HepG2 were cultured in Dulbecco Modified Eagle Medium (DMEM). All media contained 10% fetal bovine serum (Gibco, Grand Island, NY, USA) and 1 × antibiotic-antimycotic solution (Corning Inc., New York, NY, USA) and were incubated at 37 °C in 5% CO_2_.

### 4.2. Plant Materials

The plant extract (014–094) used in this research was obtained from the Korea Plant Extract Bank at the Korea Research Institute of Bioscience and Biotechnology (Daejeon, Korea). The plant was collected from Geoje-si, Gyeongsangnam-do, KOREA in 2002. A voucher specimen (KRIB 0000057) is kept in the herbarium of the Korea Research Institute of Bioscience and Biotechnology (Ochang, Korea). The plant (34 g), dried in the shade and powdered, was added to 1 L of methyl alcohol 99.9 % (HPLC grade) and extracted through 30 cycles (40 KHz, 1500 W, 15 min ultrasonication–120 min standing per cycle) at room temperature using an ultrasonic extractor (SDN-900H, SD-ULTRASONIC Co. Ltd, Seoul, Korea). After filtration and drying under reduced pressure, the *O. obtriangulata* extract (4.0 g) was obtained. The dried sample (voucher specimen: 014–094) was dissolved in dimethyl sulfoxide for experimentation. All plant materials were deposited in the Plant Extract Bank of Korea Research Institute of Bioscience and Biotechnology (KRIBB) in Daejeon, Korea (Daejeon, Korea, http://extract.kribb.re.kr/).

### 4.3. MTT Assay (Cell Viability Assay)

BxPC3 cells were seeded at 1 × 104 cells per well into 96-well plates and incubated overnight; then, they were treated with OOE (12.5–400 µg/mL) for 24 h to confirm the cytotoxicity of OOE. MTT (3-[4,5-dimethyl-2-thiazolyl]-2,5-diphenyl-[2H]-tetrazolium bromide) was added to the wells at a final concentration of 0.5 mg/mL and incubated for 2 h. The cell medium was removed, and formazan crystals were dissolved with DMSO to check absorbance. The plates were incubated for 5 min in a slow shaker, and the resulting formazan was detected using a microplate reader at a wavelength of 540 nm.

### 4.4. Colony Formation Assay

For the colony formation assay, 5000 cells were seeded in 6-well plates overnight. The medium was removed and replaced with fresh media with or without OOE once every three days for 14 days. When colonies were visible without a microscope, they were fixed with 4% paraformaldehyde and stained using 0.1% crystal violet solution. After observation, stained colonies were dissolved in DMSO, and the intensity was measured by a microplate reader at 540 nm [25].

### 4.5. Immunofluorescence Assay

The cells were treated with or without OOE in 4-well confocal dishes. After treatment, cells were fixed with 10% formalin for 10 min, treated with 0.2% Triton X in phosphate-buffered saline (PBS) for 20 min, washed three times with PBS, and then blocked with 5% bovine serum albumin (BSA) solution in PBS for 1 h. Cells were incubated overnight with anti-pSTAT3, Ki67 antibody (1:500 in PBS) at 4 °C followed by Alexa-Fluor-488 antibody and Alexa Flour-594 antibody for 1 h at room temperature. Cell nuclei were stained with 4,6-diamidino-2-phenylindole (DAPI; Sigma-Aldrich, St. Louis, MO, USA) for 3 min at room temperature. Cells stained with DAPI were washed with PBS for 30 min, and fluorescence was observed using a microscope.

### 4.6. Annexin V/Propidium Iodide Staining Assay and Analysis of Cell Cycle Distribution

Cells were seeded at 50 × 104 cells per well into 6-well plates and then treated with OOE for 24 h. To confirm apoptosis, cell cycle analysis was conducted to determine cell accumulation in the G2/M phase as described previously using flow cytometry [26].

### 4.7. Immunoblotting

Cells were completely lysed using cell lysis buffer (Cell Signaling Technology, Danvers, MA, USA). Cytosolic and nuclear fractions were divided using NE-PER Nuclear and Cytoplasmic Extraction Reagents (Thermo Fisher Scientific, Waltham, MA, USA). Bio-Rad Protein Assay Reagent (Bio-Rad, Hercules, CA, USA) was used to measure protein concentrations for the Bradford assay. Each sample was separated in sodium dodecyl sulfate polyacrylamide gel electrophoresis (SDS-PAGE) gel and then electro-transferred to a nitrocellulose membrane. The membranes were blocked with 3% BSA in tris-buffered saline/Tween 20 (TBS-T). Membranes were incubated at 4 °C overnight with the following antibodies: *p*-Src, c-Src, *p*-STAT3, T-STAT3, PARP, cleaved PARP, pro-caspase-3, cleaved caspase-3, *p*-ERK1/2, ERK1/2, Ki67, Lamin b1, and beta-actin, c-Myc, and P27 (All were 1:3000). Membranes were extensively washed with TBS-T for 1 h and then incubated with the following secondary antibodies: goat anti-rabbit IgG-HRP and goat anti-mouse IgG-HRP (all were 1:5000 in TBS-T).

### 4.8. RT-qPCR

Cells were seeded at 50 × 104 cells per well into 6-well plates and incubated with or without OOE for 24 h. Then, cells were treated with Ribo-Ex to extract RNA using the GeneAll Hybrid-R RNA Purification Kit (GeneAll, Seoul, Korea). RNA was quantified by Nano Drop (Thermo Fisher Scientific, Waltham, MA, USA). Amplification of cDNA was performed as follows: 45 °C for 60 min and 95 °C for 15 min at 4 °C using the Maxime RT premix (iNtRON Biotechnology, South Korea). Real-time quantitative PCR was performed using the Universal SYBR Green Master Mix (Applied Biosystems, Foster City, CA, USA). Real-time PCR was performed on an Applied Biosystems Step One System (Applied Biosystems, Foster City, CA, USA). In this study, relative target gene expression was quantified relative to that of GAPDH to determine mRNA expression levels. The used PCR primers are as follows:
CDK1: Forward; 5TGGAGAAGGTACCTATGGAGTT3, Reverse; 5AGGAACCCCTTCCTCTTCAC3CDK2: Forward; 5AAAGCCAGAAACAAGTTGACG3, Reverse; 5GAGATCTCTCGGATGGCAGT3VEGF: Forward; 5GGAGTGTGTGCACGAGTC3, Reverse; 5GGTCGACTGAGAGCT3Cyclin B1: Forward; 5GAACAACTGCAGGCCAAAAT3, Reverse; 5CACTGGCACCAGCATAGG3Survivin: Forward; 5TTCTGCACATCTGAGTCG3, Reverse; 5TGTCGAGAGCTCAGT3MMP9: Forward; 5TTGACAGCGACAGAGTG3, Reverse; 5GCATTCACGTCGTCCTTAT3

### 4.9. Liquid Chromatography-Mass Spectrometry (LC-MS)

Chromatographic separation of the extract was performed using an Agilent 1290 Infinity LC System (Agilent Technologies Santa Clara, CA, USA) with a Walters C18 column (2.1 mm × 100 mm, 1.7 μm) at 30 °C, which employed a mobile phase comprising 0.1% formic acid in water (A) and 0.1% formic acid in acetonitrile (B). The gradient was programmed as follows: 0–15 min, 5–95% B in A; 15–20 min, 95% B; and 20–25 min, 95–5% B; and the column was equilibrium with 5% B. A sample of 1 μL was injected into the column using an autosampler. The HPLC system was interfaced to the MS system, an Agilent 6550 Accurate-Mass Q-TOF (Agilent Technologies, Santa Clara, CA, USA) equipped with a dual agilent jet stream technology electrospray ionization (AJS ESI )source operating in positive and negative ion modes. The ESI spray voltage was set to 3500 V (Vcap). Mass spectra were acquired at a scan rate of 1 spectra/s with a mass range of 100–1700 m/z. Data analysis was performed using the Mass Hunter Qualitative Analysis Software (version B.07.00, Agilent Technologies, Santa Clara, CA, USA) for compound profiling [27].

### 4.10. Statistical Analysis

Data were expressed as mean ± SEM of at least different three experiments. Statistical analysis was performed using one-way analysis of variance in Graph Pad Prism 5, followed by Dunnett’s multiple comparison test. A *p*-value < 0.05 was considered statistically significant, and *p* < 0.01 and *p* < 0.001 were considered highly significant.

## 5. Conclusions

In this study, we investigated the inhibitory effect of OOE on the pancreatic cancer cell line BxPC3. It was demonstrated that OOE showed anti-proliferative effects and induced cell cycle arrest and apoptosis based on the inhibition of ERK/Src/STAT3 pathways and various STAT3-targeted factors. Altogether, we provide novel insights regarding OOE as a potential natural source for the suppression of pancreatic cancer. However, further component analysis is needed to achieve stronger therapeutic effects through identifying individual compounds in OOE.

## Figures and Tables

**Figure 1 molecules-25-02301-f001:**
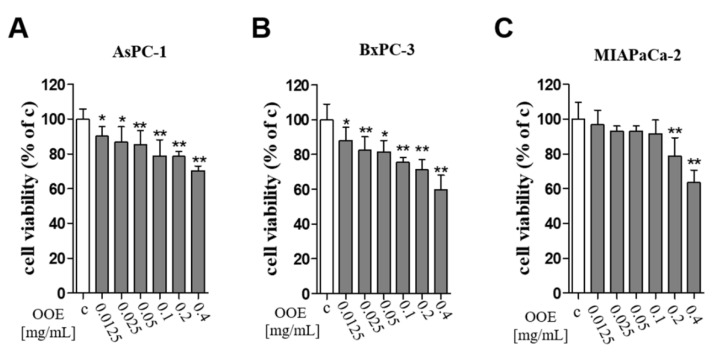
Effects of *O. obtriangulata* methanol extract (OOE) on BxPC3, AsPC-1, and MIAPACA-2 cell viability. Cells were seeded in 96-well plates and treated with OOE (0, 12.5, 25, 50, 100, 200, and 400 μg/mL) for 24 h. Cell viability was determined using MTT solution. The relative cell viability is shown as a bar graph compared with the control group (100%). MTT data are expressed as the mean ± S.D.* *p* < 0.05, ** *p* < 0.01 in comparison to the control.

**Figure 2 molecules-25-02301-f002:**
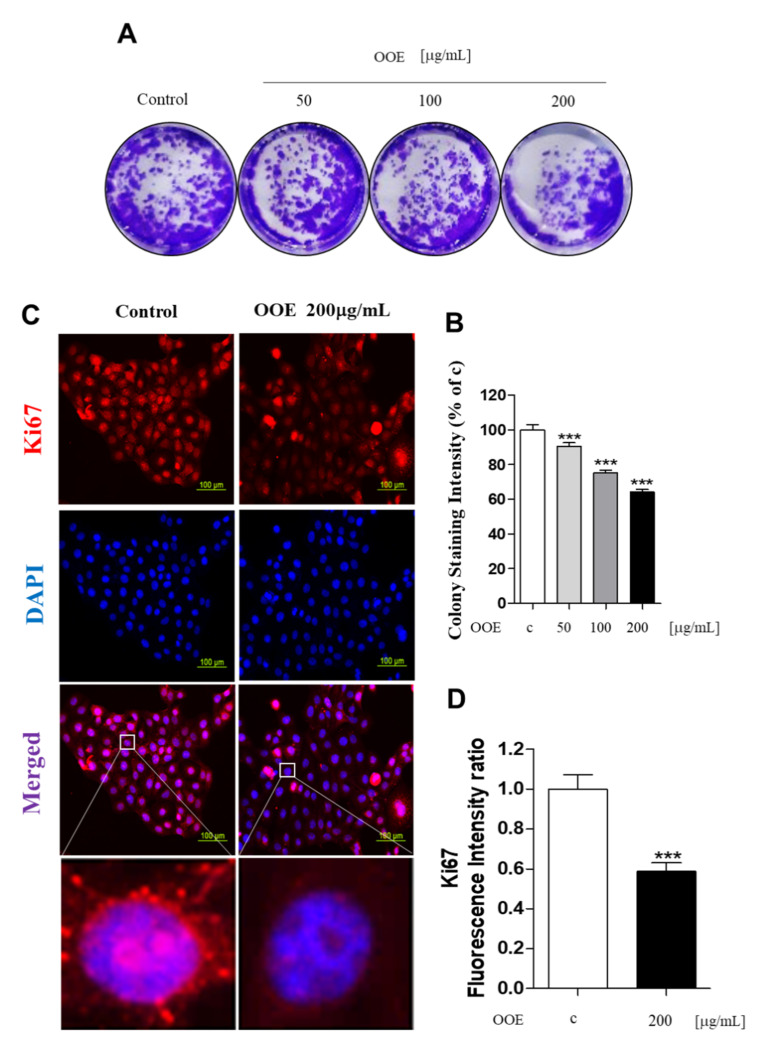
Effects of OOE on proliferation of BxPC3 cells. (**A**,**B**) BxPC3 cells were seeded in 6-well plates and incubated for 14 days in media with or without OOE (50, 100, and 200 μg/mL), which was changed every three days. Using crystal violet solution, the effect of OOE on long-term cell proliferation was detected. Formed colonies were dissolved in DMSO and shown as a bar graph compared to the control. (**C**,**D**) BxPC3 cells were seeded with or without OOE (0 or 200 μg/mL) in 4-well plates and incubated for 24 h. Cell were fixed and stained with the anti-Ki67 antibody. Stained cells were observed by a microscope (magnification: 200×, scale bar: 100 μm). Colony formation graph and fluorescence intensity are expressed as the mean ± S.D. *** *p* < 0.001 in comparison to the control.

**Figure 3 molecules-25-02301-f003:**
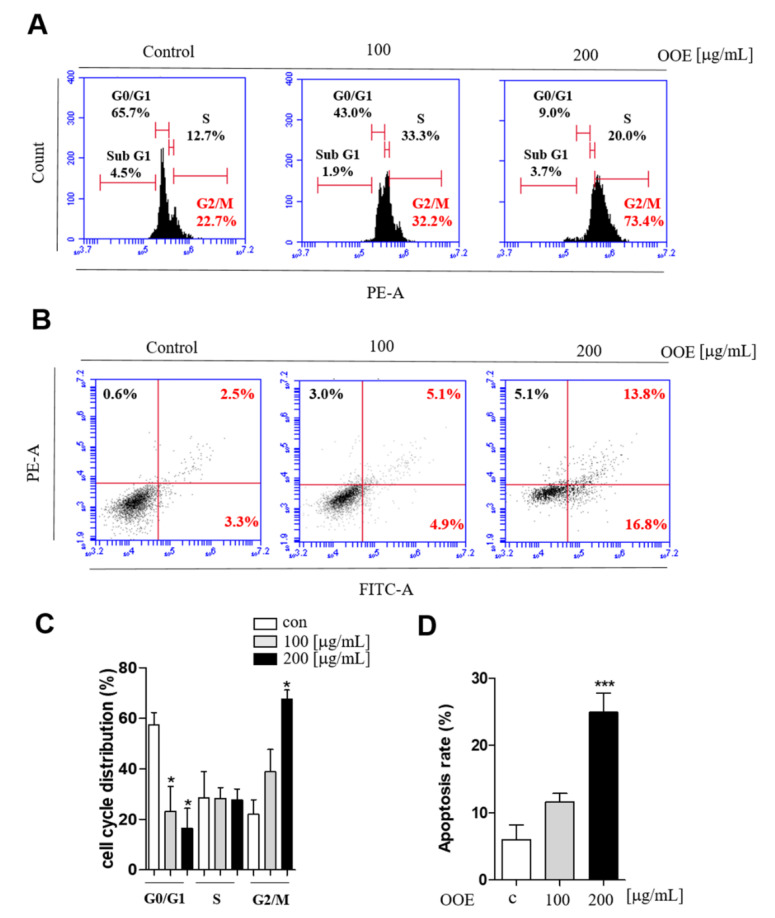
Effect of OOE on apoptosis and cell cycle arrest of BxPC3 cells. BxPC3 cells were seeded in 6-well plates and incubated with 0, 100, and 200 μg/mL OOE for 24 h. (**A**) Propidium iodide staining analysis of BxPC3 cells was performed by FACS. (**B**) Annexin V analysis was used to confirm the apoptotic effect of OOE on BxPC3 cells. (**C**) Data are presented as the mean ± SEM of three separate experiments. (**D**) The percentages of Annexin-V+/PI− (lower right quadrant) and Annexin-V+/PI+ (upper right quadrant) cells were calculated and shown as a bar graph.* *p* < 0.5, *** *p* < 0.001 in comparison to the control.

**Figure 4 molecules-25-02301-f004:**
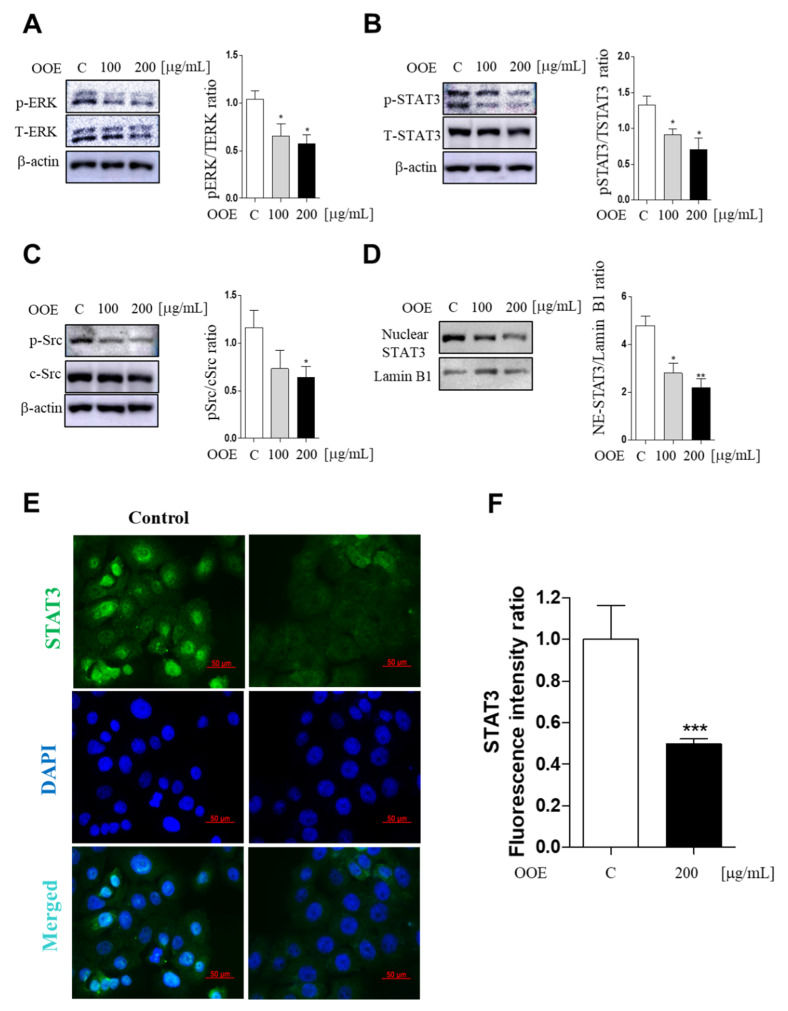
Inhibitory effect of OOE on phosphorylation of the ERK/Src/STAT3 pathway in BxPC3 cells. BxPC3 cells were treated with 0, 100, and 200 μg/mL OOE for 3 h. Cells were harvested and extracted by cell lysis buffer. Protein levels of (**A**) pERK, TERK, (**B**) *p*-STAT3, t-STAT3, (**C**) *p*-Src, c-Src, (**D**) nuclear STAT3, and Lamin B1 were detected by western blot and the expression ratio of protein phosphorylation levels were measured by analyzing immunoblots of phosphorylated proteins and total proteins. (**E**) Nuclear expression of STAT3 was confirmed by immunofluorescence assay (magnification: 400×, scale bar: 50 μm). (**F**) The fluorescence intensity was calculated by the ImageJ program and expressed as the mean ± S.D. * *p* < 0.5, ** *p* < 0.1, *** *p* < 0.001 in comparison to the control.

**Figure 5 molecules-25-02301-f005:**
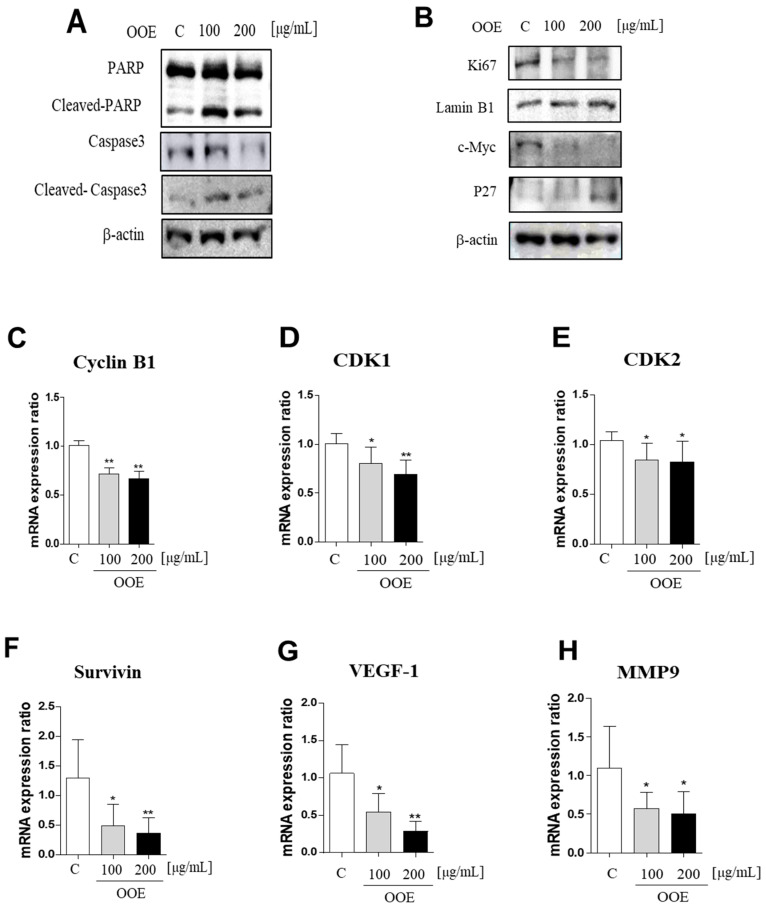
Inhibitory effects of OOE on STAT3-downregulated gene expression level cells were treated with 0, 100, and 200 μg/mL OOE for 24 h and then harvested and analyzed by western blot. (**A**,**B**) The expression levels of PARP, cleaved-PARP, caspase-3, cleaved-caspase-3, Ki67, c-Myc, p27, actin, and Lamin B1 were detected by specific antibodies. (**C**–**H**) The relative mRNA expression levels of cyclin B1, CDK1, and CDK2, which are related to cell cycle arrest, and Survivin, VEGF-1, and MMP-9, which contribute to cell survival, were confirmed by reverse-transcription quantitative polymerase chain reaction (q-PCR). The mRNA expression data were normalized with Glyceraldehyde 3-phosphate dehydrogenase (GAPDH). * *p* < 0.5, ** *p* < 0.1 in comparison to the control.

**Figure 6 molecules-25-02301-f006:**
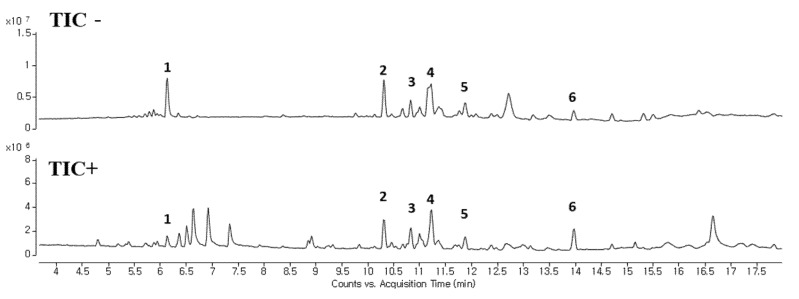
LC-MS analysis of OOE. LC-MS was performed to confirm the separation of components in OOE. A total ion chromatogram of OOE is described. Compounds with potential as active substance were makred with peak numbers (1–6).

**Table 1 molecules-25-02301-t001:** Description of the peak lists in OOE by LC-MS.

		Measured	Observed	
No.	RT	*m*/*z*	Formula	*m*/*z*
1	6.1	756	[M + H]^+^	757.2014
			[M − H]^−^	755.1952
2	10.3	676	[M + H]^+^, [M + Na]^+^	677.3747, 699.3599
			[M − H]^−^, [M − H + FA]^−^	675.3697, 721.3764
3	10.8	602	[M + Na]^+^	625.3218
			[M − H]^−^, [M − H + FA]^−^	601.3312, 647.3383
4	11.2	536	[M + H]^+^	537.3063
		560, 578	[M − H]^−^	559.3206, 577.2777
5	11.8	278	[M + H]^+^	279.2335
			[M − H]^−^	277.2201
6	13.9	278	[M + H]^+^	279.2335
			[M − H]^−^	277.2201
7	14.8	280	[M + H]^+^	281.2487
			[M − H]^−^	279.2375

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
