# Peer review of "Anti-Cancer Potential of Oxialis obtriangulata in Pancreatic Cancer Cell through Regulation of the ERK/Src/STAT3-Mediated Pathway"

_molecules, 2020, doi:10.3390/molecules25102301_

Round 1
Reviewer 1 Report
The authors have investigated that anti-cancer effect of Oxialis obtriangulata (OOE), one of natural plants in Asia and this extract promote cancer cell death via which mechanisms against human pancreatic cancer cell. Have you experimented only one pancreatic cancer cell line? The data and discussion using two or three cell lines would be enhanced your studies to reveal anti-cancer effect of OOE. The manuscript is well written, and the experiments are almost fine. However, additional explanation and some error correction are needed.
Comment:
- If you have any more data using other pancreatic cancer cell lines, please show the data and discuss about them.
- Line89-91 (Figure 1B, 1C): Authors explained “control” was none of treatment in both MTT assay and colony formation assay. Solvent control (ex; DMSO alone) is generally setup as a negative control in cell-based assay. How about effect by solvent control?
- Line110-112: Figure numbers were incorrect. Figure 2B (incorrect)--- 2C (correct), Figure 2C, D (incorrect)--- 2B, D(correct)
- Line113-114: Authors said that apoptosis was induced in only a few additional BxPC3 cell in this manuscript (Figure 2B, 2D). On the other hand, in the figure 4A, cleaved PARP and cleaved caspase3 which are typical apoptosis markers, were increased. It is apparent discrepancy between these two results. In figure 2B and 2D, apoptosis rate is total Annexin V positive cells, and little percentage (under 5%) is not meaning “apoptosis”. Whether biological mechanisms induced cancer cell death in this case was used cell cycle arrest with or without apoptosis? Re-examination or revised all your data were requested now.
- Line128-133: There is no explanation about p-ERK decreasing. Figure number (B to E) in Fig 3 was plenty incorrect.
- Line132: Figure 3F is not to be found anywhere.
- In figure 4C to 4H, why didn’t you show the protein level expression of the cell proliferation and cell cycle arrest-related molecules?
Author Response
Reviewer 1
If you have any more data using other pancreatic cancer cell lines, please show the data and discuss about them.
Before selecting BxPC3 cells, we evaluated the toxicity of OOE against three types of pancreatic cancer cells (AsPC-1, MIAPACA-2 and BxPC3) by detecting cell viability using MTT assay. As a result, in MIAPACA-2 cells, OOE didn’t showed effective inhibitions at low concentration. Cell viability of MIAPACA-2 decreased only at the high concentrations (0.2-0.4mg/mL), and the inhibitory effect was not great overall. In the case of AsPC-1, cell viability was more decreased than that of MIAPACA-2 by OOE. However, the inhibition in AsPC-1 was less than that of BxPC3 at the highest dose. Based on these results, we confirmed that OOE was most effective to decrease cell viability in BxPC3 cell among them. Thus, we experimented with BxPC3 as a main target. This is mentioned in Figure 1 and Result 2.1.
Line89-91 (Figure 1B, 1C): Authors explained “control” was none of treatment in both MTT assay and colony formation assay. Solvent control (ex; DMSO alone) is generally setup as a negative control in cell-based assay. How about effect by solvent control?
We’re sorry that we misrepresent the experimental meaning of control. Actually, all assay including MTT (not cell culture) were performed using media that contained 0.5% DMSO. So ‘control’ also contained 0.5% DMSO. To avoid misunderstanding, we deleted all part which were written as ‘non-treated’. Thanks for the detailed advice.
Line110-112: Figure numbers were incorrect. Figure 2B (incorrect)--- 2C (correct), Figure 2C, D (incorrect)--- 2B, D(correct)
The composition of our figures has changed slightly for revision. According to the detailed reviewer’s advice, the figure number and result were corrected and entered.
Line113-114: Authors said that apoptosis was induced in only a few additional BxPC3 cell in this manuscript (Figure 2B, 2D). On the other hand, in the figure 4A, cleaved PARP and cleaved caspase3 which are typical apoptosis markers, were increased. It is apparent discrepancy between these two results. In figure 2B and 2D, apoptosis rate is total Annexin V positive cells, and little percentage (under 5%) is not meaning “apoptosis”. Whether biological mechanisms induced cancer cell death in this case was used cell cycle arrest with or without apoptosis? Re-examination or revised all your data were requested now.
We accepted that reviewer’s opinion that when the rate of apoptosis was less than 5%, that it was difficult to see it has an apoptotic effect, then we re-tested. We were also puzzled by the few effects compared to western blot data (PARP, Caspase-3). By improving the error of our first FACS assay, we were able to obtain better data consistent with the protein data. As a result of the re-examination, similar to the PARP and caspase protein data, the early and late apoptosis rated increased to more than 10 %. We changed the Figure 3B and D with the modified FACS data.
Line128-133: There is no explanation about p-ERK decreasing. Figure number (B to E) in Fig 3 was plenty incorrect.
The reduction of pERK was mentioned in line 143 with reduction of Src and STAT3.
Line132: Figure 3F is not to be found anywhere.
We removed the figure numbers that weren’t showed and fixed them all to the correct location.
In figure 4C to 4H, why didn’t you show the protein level expression of the cell proliferation and cell cycle arrest-related molecules?
We agreed with the reviewer’s advice to enhance the validity of our research and conducted additional western blot. We showed additional two factors (p27 and c-Myc) related to cell cycle and proliferation in Figure 5B. c-Myc is a protein involved in the cell cycle, which is overexpressed in most of cancer types, stimulating the cell cycle process and helping cancer cell grow well. The c-Myc targets and regulates the p27, tumor suppressor gene. p27 inhibits activity of cyclin-cdk complex by binding them suppressing the cell growth and proliferation. In our western blot data, we found OOE reduced the expression of c-Myc (oncogenic protein) and increased the expression of p27(tumor suppressor protein). Thus, we support cell cycle arrest data (FACS) with western blot data by showing p27(cell cycle, proliferation), Ki67(most popular proliferation marker), c-Myc(cell cycle marker) expression regulated by OOE.
Thanks for your detailed advice.
Reviewer 2 Report
In this manuscript, the authors tested the anticancer activities of O. obtriangulata methanol extract (OOE) in a pancreatic cancer cell line and proposed the mechanism of action by inhibiting STATA3 pathways. Although this manuscript might be interesting in natural products field, lacking proper controls and convincing data to support the conclusion seriously impair the quality of this work. Therefore, it is not recommended to publish on Molecules.
My specific major concerns are:
- The authors used the dried sample OOE in DMSO. However, the negative control is no treatment. The authors should add DMSO negative control in their assays (Fig. 1-4).
- The authors claimed that OOE can be an alternative therapy to overcome the toxicity and drug resistance problem by chemotherapy and radiotherapy, but showed no data. The authors should include at least one control cell line and monitor the toxicity of OOE to see whether OOE is selectively toxic to cancer cells.
- OOE is a mixture of several compounds, and they probably target different genes, making it difficult to link the anti-proliferative activity to inhibiting only one type of pathways, STAT3-related pathways. The authors need additional experiments to elucidate the mechanism of action.
1) It is suggested to isolate each compound in OOE and test their activity individually.
2) The authors should use siRNAs to knock down STAT3 related genes or CRISPR to knock out STAT3 related genes to see whether the cytotoxicity is mitigated.
- The authors should quantify the immunofluorescence assays in Fig. 1D and Fig. 3E.
- Fig. 1-2 showed the cytotoxicity of OOE to BxPC3 cells is mild. What’s the potency of OOE compared to approved chemotherapy drugs?
Some minor points:
- Page 6, line 132, “As shown in Figure 3F” should be “Fig. 3E”.
Author Response
Reviewer 2
The authors used the dried sample OOE in DMSO. However, the negative control is no treatment. The authors should add DMSO negative control in their assays (Fig. 1-4).
We misrepresent the experimental meaning of control. Actually, all assay including MTT (not cell culture) were performed using media that contained 0.5% DMSO. So ‘control’ also contained 0.5% DMSO. To avoid misunderstanding, we deleted all part which were written as ‘non-treated’.
The authors claimed that OOE can be an alternative therapy to overcome the toxicity and drug resistance problem by chemotherapy and radiotherapy, but showed no data. The authors should include at least one control cell line and monitor the toxicity of OOE to see whether OOE is selectively toxic to cancer cells.
Alternative therapy can be applied as a way to supplement the shortcomings of general chemotherapy or to increase their efficacy. However, the main purpose of our study is the discovery of the natural material Oxalis which has anti-cancer efficacy against pancreatic cancer cell not alternative therapy. We intended to mention the OOE’s anti-cancer efficacy and “possibility” of suppression of ERK which can be activated by Gemcitabine in BxPC3 rather than “toxicity” or “alternative therapy”.
In the discussion part, we found that the meaning of data was incorrectly written. So, we understand ‘why the reviewer focused on the activity of OOE as an alternative therapy’. Thus, to modify the meaning, we changed ‘but also potentially lowers~’ to ‘but also has a possibility to lower~’ in line 222. Additionally, the word ‘resistance’ was changed to specifically ‘activated ERK’.
And in fact, for this revision, we conducted additional experiments to confirm that OOE can reduce ERK activated by Gemcitabine. As a result, when 10mM gemcitabine was treated for 24 hours on BxPC3, we confirmed that ERK can be overexpressed by Gemcitabine. And OOE treated cells showed decreased levels of p-ERK. This data is attached to Supplementary Figure 2. According to this result, OOE can decrease the expression level of ERK increased by Gemcitabine in BxPC3 cells.
Additionally, in fact, we can not claim that the OOE only work selectively on cancer cells. This paper focused on the anti-cancer potential of the new natural substances. According to the reviewer’s opinion, we studied with normal cells. Unfortunately, we’re sorry that there are no normal pancreatic cell lines we can buy now. Instead, we performed MTT using our gastric normal cell, GES-1. As a result, the cell viability was higher in GES-1 than in BxPC3 when cells are treated with OOE for 24hours. OOE didn’t showed high toxicity in GES-1(normal cell) compared to BxPC3.
This result dose not prove that OOE selectively acts only on cancer cell, however, as shown in Figure 1 and Sup. Figure1, OOE may have different inhibitory effects for each cell. Furthermore, Oxalis family has been used as a natural remedy based on anti-inflammatory effects in Asia. we speculated that OOE is believed to be difficult to see as having storing fatal toxicity to human.
OOE is a mixture of several compounds, and they probably target different genes, making it difficult to link the anti-proliferative activity to inhibiting only one type of pathways, STAT3-related pathways. The authors need additional experiments to elucidate the mechanism of action.
1) It is suggested to isolate each compound in OOE and test their activity individually.
We also agree, as the reviewer’s comment, that compound isolation allows us to know which substances represent the anti-cancer potential of OOE. We conducted LC-MS to investigate the ingredients. However, the exact ID(identification) of compounds usually need to match LC-MS data to previous other analysis research papers. The important point is ‘Matching’ with other reference articles. Unfortunately, analytical studies using OOE have not been conducted, and our paper is the first anti-cancer study using this extract.
Accordingly, there is no reference material to match with our data. Moreover, Identification of unknown compounds is a task that requires additional analysis equipment such as NMR, analytical experts, and a very long time. For this reason, the ID of compounds in OOE is not possible.
However, we tried to add as much information as possible in Table 1. To provide information, we showed which molecules or elements (H, Na, FA-formic acid.etc) the predicted molecular weight was calculated from. And we added the calculation formula in Table1. Although our paper did not provide the exact compositions in OOE, we expect this to be the cornerstone for further analysis research.
2) The authors should use siRNAs to knock down STAT3 related genes or CRISPR to knock out STAT3 related genes to see whether the cytotoxicity is mitigated.
Among the STAT family, STAT3 is most closely related to tumorigenesis and is well known as oncogene. Accumulating evidence strongly implicates the critical role of aberrant STAT3 activation in malignant transformation and tumorigenesis. The contribution of STAT3 to cancer growth is already established, and there have been papers reporting that cell growth has progressed very slowly by silencing STAT3.
According to this report (Nature Reviews Cancer volume 14, pages 736–746(2014)), STAT3 pathway plays a crucial role in cancer cell proliferation, survival, tumor immunosuppression, angiogenesis and metastasis. The importance of this signals is not limited to cancer and extend to cancer stem-like cells. Meanwhile, STAT3 is closely related to cell growth. In human nervous cancer (neuroblastoma), the deletion of STAT3 induced apoptosis compared with vector suppressing the expression of Bcl-xl. It is confirmed by Annexin-V/FITC assay and western blot assay. And using MTS assay, negative mutant- STAT3 group showed very low growth rate compared with vector group (Oncogene volume 21, pages 8404–8413(2002)). In another study (International Journal of Biological Macromolecules, Volume 149, 15 April 2020, Pages 487-500), it was confirmed that silencing STAT3 made cells grow poorly compared to control in 4T1, CT26 and B16-F10 cells. the mRNA levels of BIM and Bcl-2 induced in si-STAT3 group. Additionally, si-STAT3 suppressed colony formation of SK-BR-3(human breast cancer cell) and inhibited the migration of SK-BR-3 cell too (Tropical Journal of Pharmaceutical Research September 2018; 17 (9): 1753-1758). As explained above, in many research papers, effects such as growth inhibition, apoptosis induction and migration inhibition have been reported in STAT3-silenced cells. Due to the growth inhibitory effect of si-STAT3, the cell viability can be low in the results of cytotoxicity test such as MTT, MTS.
The authors should quantify the immunofluorescence assays in Fig. 1D and Fig. 3E.
According to the reviewer’s advice, we quantified the fluorescence intensity using image J program. In the case of Ki67, about 40% of red fluorescence was reduced in OOE-treated group. It was inserted into Figure 2D as a bar graph. In the case of STAT3, OOE-treated group showed approximately 60% reduced green fluorescence compared to control group. It was also inserted in Figure 4B. Thank you for your careful advice to increase to validity of the data.
Fig. 1-2 showed the cytotoxicity of OOE to BxPC3 cells is mild. What’s the potency of OOE compared to approved chemotherapy drugs?
In considering drug selection, we considered the OOE’s cytotoxicity and inhibitory effect on factors (ex. STAT3) together on BxPC3. In case of Gemcitabine, it acts as dFdC to prevent the synthesis of DNA and has the effect of inhibiting cell growth, In the case of OOE, we used concentration that were not very toxic, but significantly decrease the cell viability. And in this additional MTT study, it was confirmed that the cytotoxicity of OOE on BxPC3 was greater than that of other cells. And OOE showed significant reduction of ERK, Src and STAT3 proteins. STAT3, transcriptional factor, is the one of the potential drug target for cancer therapy. Because STAT3 mediates cell growth, differentiation and survival in cancer as a major intrinsic pathway for cancer. And OOE can regulate the activity of STAT3 and then can have an anti-proliferative, cell cycle arrest, apoptotic effect. Considering that it has been used for a long time as a medicinal plant based on anti-inflammatory and hemostatic action, safety can be guessed. Overall, we found the effects of anti-proliferation, cell cycle blockage and apoptosis induction of OOE, mediating ERK/Src/STAT3 in BxPC3. We present new natural materials with anti-pancreatic cancer efficacy.
Page 6, line 132, “As shown in Figure 3F” should be “Fig. 3E”.
According to the detailed reviewer’s advice, the figure number and result were corrected and entered.
Thanks for your detailed advice.
Reviewer 3 Report
Authors should explain why pancreatic cancer has been chosen, and what might be the effect of the extract on e.g. lung cancer or other cell lines.
Authors should mention some components of the OOE, if already descibed in the literature.
Authors should explain all abbreviations.
Authors should explain why an ~5% apoptosis rate is significant. Moreover, what is the effect of the extract on non-cancer cells?
Authors are suggested to measure high resolution MS in order to characterize the components of the extract in a more proper ways, e.g. chemical formula.
The extraction protocol should be descibed in more details.
Author Response
Reviewer 3
Authors should explain why pancreatic cancer has been chosen, and what might be the effect of the extract on e.g. lung cancer or other cell lines.
The reason why we focus on pancreatic cancer is that pancreatic cancer is one of the most difficult to diagnose and has a poor prognosis, so our lab performed finding natural source with anti-cancer potential usually targeting pancreatic cancer cells. ‘phytomedicine, 56 (2019)48-56, Kim Yumi’ is our recent study using pancreatic cancer. According to the reviewer’s opinion, we attached more MTT data using A549 (human lung cancer) and HepG2(human liver cancer) cells in supplementary data. As a result, in HepG2, OOE very slightly suppressed cell viability. It didn’t show big toxicity on HepG2. In the case of A549, cell survival was decreased in the concentration-dependent manner by OOE, but the survival rated was over 60% even at the highest concentration(0.4mg/mL). Moreover, we added additional MTT data using pancreatic cell lines (MIAPACA-2, AsPC-1 and BxPC3) in Figure 1. As a result, among three pancreatic cancers, in BxPC3, OOE showed most powerful inhibition. That’s why we first select ‘BxPC3’ to our research. And this content was added to result 2.1.
Authors should mention some components of the OOE, if already described in the literature.
Oxalis family is generally reported to have components such as oxalic acid, malic acid and tartaric acid. We added this briefly in the introduction part, line77.
Authors should explain all abbreviations.
Words without abbreviation explanations have been added to the abstract with additional explanation.
For examples, extracellular-signal-regulated kinase (ERK), Proto-oncogene tyrosine-protein kinase Src (Src), Matrix metallo peptidase9 (MMP9), Vascular endothelial growth factor-1 (VEGF-1), Cyclin-dependent kinase inhibitor 1B (p27).
Authors should explain why an ~5% apoptosis rate is significant. Moreover, what is the effect of the extract on non-cancer cells?
As a result of our FACS data, it was judged to be significant in that is increase repeatedly even if the increase was very small. However, considering the western blot data, we decided that we could retest and get a better data improving the errors of our protocols. As a result, we could observe the apoptosis rate to more than 10%. The modified FACS data are attached to Figure 3B and D.
Unfortunately, there was no pancreatic normal cell that we could purchase now. we’re sorry that we couldn’t make a contrast properly. But we conducted MTT assay using human gastric normal cell line, GES-1, the only normal cell in our lab, to accommodate the opinions of reviewer as much as possible.
As a result, OOE affect cell viability of GES-1, but the inhibition was much less than BxPC3. We can not say OOE have any effect on the normal cell line. However, we can say that the toxicity of OOE may work differently depending on the cell types. Because in figure1 and Sup.figure 1, OOE showed different inhibition rate of viability on different 6 cell lines (BxPC3, MIAPACA2, AsPC1-pancreatic cancer, A549-Lung cancer, HepG2-liver cancer and GES1-gastric normal cell). Additionally, since Oxalis family has been used for a long time in Asia as an herbal medicine (anti-inflammatory agent, etc.), it is difficult to assume that it is fatally harmful to human body.
Authors are suggested to measure high resolution MS in order to characterize the components of the extract in a more proper ways, e.g. chemical formula.
In fact, the MS we used has good quality enough to suggest quite clear molecular weight. However, in order to identification of components in OOE, we need to match our data with previous analysis paper about OOE. The key point is ‘matching’ with other many reports. However, there is no prior analysis data of OOE and maybe our LC-MS data is the first report about the component of OOE, so it is difficult to match. Moreover, the process of identifying unknown components is very difficult requiring advanced analysis equipment as NMR and taking a long time. That’s why we suggest only expected mass value. We really sorry that we can’t give a chemical formula because we don’t know and can’t expect any of component. However, to provide as much as possible, we suggest which molecules or elements (H, Na, FA-formic acid,etc.) the predicted molecular weight was calculated from. And we added the calculation formula in Table 1.
The extraction protocol should be described in more details.
The extract we used were purchased from KRIBB as mentioned in Materials and Method section. We asked KRIBB to provide more detailed information for reflecting reviewer’s idea. More details including (the sampling time, area, solvent, yield, etc.) were written in Materials & Methods 4.2 section.
Thanks for your kind and detailed advice.
Round 2
Reviewer 1 Report
The authors have investigated that anti-cancer effect of Oxialis obtriangulata, one of natural plants using as an herbal medicine in Asia and this extract promote cancer cell death via which mechanisms against human pancreatic cancer cell. The authors answered to all comments with sincerely and added the accurate new data and explanation. However, the data are partially insufficient. Thus, the minor revision is needed.
Comments:
In Figure 3B and D, authors tried to re-exam and showed new data by flowcytometory. After administration of OOE 200 μg/mL to BxPC3 cells, apoptotic cells were 30.6 % (13.8 + 16.8%) (Fig.3B). Would you observe the bar graph in Fig.3D carefully? In case of OOE 200 μg/mL addition, it is not seemed over 30% apoptotic rate with filled bar in this figure. Authors should affirm all numerical value, graph, and appropriate manuscript in whole.
Author Response
Dear Reviewer,
Reviewer 1
Q.
In Figure 3B and D, authors tried to re-exam and showed new data by flowcytometory. After administration of OOE 200 μg/mL to BxPC3 cells, apoptotic cells were 30.6 % (13.8 + 16.8%) (Fig.3B). Would you observe the bar graph in Fig.3D carefully? In case of OOE 200 μg/mL addition, it is not seemed over 30% apoptotic rate with filled bar in this figure. Authors should affirm all numerical value, graph, and appropriate manuscript in whole.
A.
Thanks to the reviewer's meticulous advice, we were able to complement the experimental results. Following the reviewer's advice, we found that the results of the FACS experiment were incorrectly described based on fragmented data rather than whole data. Then, based on the overall experimental results of figure 3D, the percentage average was calculated and the manuscript was revised accordingly. We added this to lines 109-114.
Reviewer 2 Report
The authors have addressed all my concerns.
Although the anti-proliferative activity might not only due to inhibiting ERK/Src/STAT3-mediated pathways, the authors showed that the anti-proliferative activity is correlated with inhibiting STATA3 pathway.
Author Response
Dear Reviewer
Reviewer 2
Q.
The authors have addressed all my concerns.
Although the anti-proliferative activity might not only due to inhibiting ERK/Src/STAT3-mediated pathways, the authors showed that the anti-proliferative activity is correlated with inhibiting STATA3 pathway.
A.
Thanks for your thoughtful concerns. Thanks to the reviewer's advice, we were able to think deeply about the experiment.
Reviewer 3 Report
Detailed answers of the authors is appreciated. There is only one more comment suggested to be included to the manuscript. Authors claim that oxalic acid, malic acid and tartaric acid are components identified earlier. However, in their LCMS chromatogram there is no mathing for these components. Authors should explain why they have not found the known components.I addition, although they claim not to be able to identify the detected compounds, they should emphasize that probably new components have been observed, and the structure of those might be interesting to be solved in the future.
What have been suggested earlier is to measure high-resolution MS (HRMS) that leads to the detection of molecular weight for more digits (e.g. 270.5678), and this usually gives information (belongs to only one corresponding structure) on the chemical formula of the compounds. This reviewer still believes that this experiment would be a great contribution to this paper.
Author Response
Dear Reviewer,
Reviewer 3
Q.
Detailed answers of the authors is appreciated. There is only one more comment suggested to be included to the manuscript. Authors claim that oxalic acid, malic acid and tartaric acid are components identified earlier. However, in their LCMS chromatogram there is no mathing for these components. Authors should explain why they have not found the known components.I addition, although they claim not to be able to identify the detected compounds, they should emphasize that probably new components have been observed, and the structure of those might be interesting to be solved in the future.
What have been suggested earlier is to measure high-resolution MS (HRMS) that leads to the detection of molecular weight for more digits (e.g. 270.5678), and this usually gives information (belongs to only one corresponding structure) on the chemical formula of the compounds. This reviewer still believes that this experiment would be a great contribution to this paper.
A.
We showed analysis data which mainly represented major peaks in Figure 6 to find the main substances which have the anti cancer-efficacy of OOE rather than the common substances (oxalic acid, malic acid and tartaric acid).
However, according to the reviewer's advice, we analyzed the existence of three acidic compounds in OOE. Although not shown in Figure 6, it was assumed that three acid ingredients exist in OOE. This was added as supplementary data 3. Since all three components are acid, it can be confirmed in the negative ion chromatogramm.
We present the calculation method for idenfying three substances and each chromatogram along with the total negative ion chromatogram in Sup.figure3.
And the reason why it dectected so early is that polaritic compounds (such as acids) are relatively less separated and quickly detected. Because under normal column separation conditions, non-polarity is gradually increased as the retection time passes. For this reason, in general, polar substances are quickly detected in front, and over time, non-polar substances are more clearly separated and detected later.
In addition, we have already shown LC-MS data using the High resolution UHPLC-QTOF 6500 and we already have a value of four decimal places. In response to the reviewer's opinion, a specific mass value (including four digits) was suggested to table 1. We hope this information will help other researchers in analytical research.